# Temperature-Dependent Sex Determination in Crocodilians and Climate Challenges

**DOI:** 10.3390/ani14132015

**Published:** 2024-07-08

**Authors:** Boglárka Mária Schilling-Tóth, Scott M. Belcher, Josefine Knotz, Silvia Ondrašovičová, Tibor Bartha, István Tóth, Attila Zsarnovszky, Dávid Sándor Kiss

**Affiliations:** 1Department of Physiology and Biochemistry, University of Veterinary Medicine, 1078 Budapest, Hungary; schilling-toth.boglarka.maria@univet.hu (B.M.S.-T.); josefine.knotz@gmail.com (J.K.); bartha.tibor@univet.hu (T.B.); toth.istvan@univet.hu (I.T.); kiss.david@univet.hu (D.S.K.); 2Department of Biological Sciences, Center for Human Health and the Environment, North Carolina State University, Raleigh, NC 27695, USA; smbelch2@ncsu.edu; 3Department of Biology and Physiology, University of Veterinary Medicine and Pharmacy in Košice, 041 81 Košice, Slovakia; silvia.ondrasovicova@uvlf.sk; 4Agribiotechnology and Precision Breeding for Food Security National Laboratory, Institute of Physiology and Nutrition, Department of Animal Physiology and Health, Hungarian University of Agricultural and Life Sciences, 7400 Kaposvár, Hungary

**Keywords:** crocodile sex determination, aromatase, anti-Müllerian hormone, SOX-9, SF-1, TRPV, miRNA, DNA methylation, climate change

## Abstract

**Simple Summary:**

The sex determination (SD) of crocodilians is a perfect example of how complex mechanisms drive evolutionary adaptation and why dramatic environmental changes endanger animals who survived millions of years on this earth. In this review, we review the mechanisms and the known influential factors of the temperature-dependent type of SD, like hormones, genes, and epigenetic modifications. Furthermore, the study considered possible, and increasing risk of climate change on the adaptation mechanisms during the nesting of crocodilians.

**Abstract:**

The sex of crocodilians is determined by the temperature to which the eggs, and hence the developing embryo are exposed during critical periods of development. Temperature-dependent sex determination is a process that occurs in all crocodilians and numerous other reptile taxa. The study of artificial incubation temperatures in different species of crocodiles and alligators has determined the specific temperature ranges that result in altered sex ratios. It has also revealed the precise temperature thresholds at which an equal number of males and females are generated, as well as the specific developmental period during which the sex of the hatchlings may be shifted. This review will examine the molecular basis of the sex-determination mechanism in crocodilians elucidated during recent decades. It will focus on the many patterns and theories associated with this process. Additionally, we will examine the consequences that arise after hatching due to changes in incubation temperatures, as well as the potential benefits and dangers of a changing climate for crocodilians who display sex determination based on temperature.

## 1. Introduction

The term “sex determination” (SD) traditionally refers to the biological process in which an organism begins the formation of either ovaries or testes from the embryonic gonad. This complex process involves multiple molecular events that guide a group of cells toward a unified tissue outcome [1].

The origins of theories on sex determination (SD) and its underlying process may be traced back to renowned thinkers like Parmenides and Aristotle [2,3]. Indeed, their hypotheses on the influence of embryo localization in the uterus or the origin of sperm from a specific testicle fail to provide a scientific understanding of the organization of embryonic tissues into either testes or ovaries [2]. The identification of essential components of the reproductive system, such as Sertoli cells or Graafian follicles, and finally the detection of sex chromosomes by Nettie Stevens, provided a head start to the detailed examination of the SD mechanism in mammals, reptilians, and amphibians [2,3]. In vertebrates, there are two main forms of sex determination: genetic sex determination (GSD) and environmental sex determination (ESD) [1,4,5,6].

This review will provide the latest TSD data, including patterns and embryonic development alterations, and discuss current TSD mechanisms. Due to the SD mechanistic variety and the evolutionary success of SD among reptiles, the distinctions between crocodilian SD and SD in other reptiles will be highlighted. In addition, we will critically examine the available data to predict how global temperature change may affect TSD species in the future.

## 2. Sex Determination in Crocodiles

Crocodilians show a distinct characteristic known as temperature-dependent sex determination (TSD), which differs significantly from the molecular basis seen in mammalian vertebrates. The main reason why this unique mechanism is dominant is because reptiles do not have specific sex chromosomes that are different in shape, form, and function [1,4,6,7,8,9,10,11].

### 2.1. Patterns of Determination

The basic definition for TSD is that different incubation temperatures promote either male or female development. TSD in crocodilians can be categorized, and the relevant temperatures during incubation can be determined. So far, three patterns of TSD have been described in reptiles, and we will refer to the patterns by TSD1a (or male to female/MF), TSD1b (or female to male/MF), and TSD2 (or female to male to female/FMF), which are also used by the authors González (2019) and Valenzuela and Lance (2004) [9,11,12,13].

These patterns are determined by the varying temperature effects that occur during the incubation stage of the embryo. Lower temperatures can result in the development of males, while higher temperatures can lead to the development of females, or vice versa. The TSD1a (MF) hypothesis proposes that male embryos hatch at lower temperatures, while female embryos hatch at higher temperatures. Although this pattern is common among turtles, it has not yet been observed in crocodilians [14]. TSD1b (FM) presents the inverse image of Ia, where cooler temperatures result in the production of females and higher temperatures result in the production of males. Based on the current understanding, there is a third distinct pattern, known as TSD2 (FMF), which is observed in specific species and is regarded as the most significant pattern when it comes to crocodiles and alligators [13,15,16]. The authors Ewert et al. (1994) described the TDS2 scheme as the “basal or ancestral” origin for TSD and further suggested that it serves as a template for the first two systems [8]. In this instance, female embryos are formed at high and low temperatures, while male embryos occur at moderate temperatures. It is also of consideration that intersexes may develop in the intermediate temperature zone [11]; however, this has not yet been proven in crocodilians [12,13,17,18,19].

As a way of determining the significance of temperatures and their effect on the embryo’s gonads, several laboratory experiments have been carried out in different species. Some examples of these studies are as follows: in the American alligator *(Alligator mississippiensis)*, temperatures below 30 to 31 °C produce female offspring exclusively. Intermediate climatic conditions result in both male and female embryos, with, in the case of *A. mississippiensis*, a strong bias towards females [18,19]. Higher temperatures above 33 °C create favorable conditions for male development, with 100% phenotypic males occurring at 33 °C [12,18,19]. The TSD patterns of the Nile crocodile, *Crocodylus niloticus*, fall within the same ranges as *A. mississippiensis*. Furthermore, the sex ratio in this species weighs more in the direction of female individuals [15,20]. Additionally, in the Estuarine crocodile, *Crocodylus porosus*, climatic conditions below 30 °C and above 34 °C exclusively promote the production of female embryos [21]. In the data obtained by Ferguson (1983) and colleagues, alligators can be classified into the TSD1b pattern, meaning females produced at low temperatures and males at higher temperatures, and recent investigations, such as González et al. (2019) [13] propose the classification of a TSD2 pattern [12,19], based on the assumption that type 1b of TSD classification may be a result of pattern TSD2, where the female-producing temperatures are too high for offspring survival [22].

Another key definition is the intervals between the female- and male-producing temperatures, the transitional ranges of temperatures (TRTs). Within these ranges, either female or male embryos can develop, leading to mixed-sex ratios [11]. Based on the different patterns of TSD, there can either be one or two TRTs. In each, TSD1a and TSD 1b, one TRT is found as there is a single interval between the transition from male to female and vice versa. Regarding TSD2, two TRTs are determinable, one between the low-temperature-female to an intermediate-male and another between the intermediate-male and high-temperature-female [11,13,16].

Concerning the width of the temperature transitions there are variations between the species. For instance, sometimes, there are intervals of almost 2 °C (mugger crocodile, *Crocodylus palustris*) in which both males and females can emerge but can be as narrow as 0.3 to 0.9 °C, as demonstrated in *A. mississippiensis* and the Morelet’s crocodile, *Crocodylus moreletii* [10,12,13].

The pivotal temperature (PT), often known as the threshold temperature, is another significant parameter in incubation experiments conducted at constant temperatures [23]. PT refers to a specific temperature that, once maintained consistently throughout incubation, can result in a 1:1 sex ratio within a set of offspring [11,22,23]. While in genotypic SD species, a 1:1 ratio of males to females is applicable and favorable, in crocodilians, PT produces a wider range of male and female ratios [24]. Not only in laboratory conditions but also in natural nest sex ratios in crocodiles and alligators are skewed and prominently biased towards a female-dominated ratio [7,12,13,16,19,20,21,24,25]. Likewise, for TRT, the number of PTs is linear with the patterns. Thus, one PT can be detected in each of TSD1a- and b and two in TSD2 [11,22]. As seen in Figure 1, two PTs are detectable in the case of *A. mississippiensis*, one at 31.7 °C, and the other at 35.9 °C [13]. Variations in PT are quite common amongst species and can be subject to different various factors. As PT does not apply to an individual egg and is regarded in the context of clutches, clutch effects can cause differences in PT of the same species. Such alterations can be quite substantial, especially in free-roaming animals and their progeny, as possible shifts could drastically change the sex ratio of a clutch [23].

Laboratory experiments, coupled with a constant and favorable climatic environment and no external factors intervening with embryonic development, provide ideal conditions to determine the above-mentioned intervals. In open nature, climatic conditions are less stable and usually fluctuate not only in Celsius degrees but also in weather conditions, such as humidity and sunlight exposure. As opposed to an artificially created incubation, free nests are more exposed during their approximately 70 to 100 days of incubation [17,19]. Depending on the environmental conditions, nest temperatures can fluctuate as much as 4 °C to 10 °C during the day [20,25].

Alternating conditions during incubation may raise the question of whether TSD patterns behave differently in naturally occurring nests. Ferguson and Joanen (1982) [19] performed studies on free clutches of *A. mississippiensis* to find answers for that obstacle and gave significant insight into TSD in wild roaming animals. Among the cooler degrees, between 29 and 31 °C, more female offspring hatched. Their eggs were mainly located in the periphery or the lower part of the clutch. Warmer temperatures around 34 °C produced mainly male individuals, which were located at the top of the nest [24]. In addition to *A. mississippiensis,* TSD was also found in crocodiles, such as *C. porosus* [21] and *C. niloticus* [20], but with some differences. Hot and cold temperatures ultimately produced females, while intermediate temperatures between 31 °C and 32 °C predominantly favored male embryos [21]. These data suggest that TSD consequently is naturally occurring and not just inducible by constant incubation temperatures provided by artificial experimental conditions.

### 2.2. Thermosensitive Period

Sex determination in crocodilians does not occur at the time of fertilization, but is set during a specific time of incubation (embryonic development) and is an irreversible organizational process with no possibility of sex reversal [24]. The developmental period of temperature sensitivity is also referred to as the thermosensitive period (TSP) [23]. It is defined as the timeframe during embryonic development where gonadal differentiation begins and where variations in incubation temperature strike an effect on the sex ratio [11,12,23,26].

The timeframe of the TSP is determined by the incubation at consistent temperatures, starting with a temperature that produces just one sex and then switching to a temperature that produces the other sex after a specific duration [4,10,16,23]. Ferguson and Joanen demonstrated the TSP effect in *A. mississippiensis*, where eggs were incubated at 30 °C, a female-producing temperature transitioned to a male-producing temperature, 34 °C, for the rest of the artificial incubation. Then, eggs were separated into groups and switched to different temperatures at alternating timestamps during their development [4]. Besides the single switches, “shift-twice” experiments are a common alternative [12] where eggs are initially switched to the opposite-sex-inducing temperature and then eventually transitioned back to the starting point, to induce sex reversal [12].

Multiple factors have been demonstrated to impact the temperature changes in crocodile embryos. In a study carried out by Deeming and Ferguson (1989) [18], it was shown that temperature transitions lasting for 7 days had a minimal impact on the background and were not enough to induce any substantial changes in the sex ratios of *A. mississippiensis*. In another investigation by Lang and Andrews (1994) on TSP [12], they further analyzed the data and found that, for the temperature switches to have a lasting effect on sex reversal, both temperature and a significant embryo size were necessary. It should be noted that this transition affects the time of the hatchling, the antipredator behavior, the feeding and thermal responses, and even growth. The transition to male caused smaller males with a lesser chance of reproduction [21,27]. It has been established that there is a direct relationship between the size and the proportion of male offspring generated during the TSP. For example, there was a shift in temperature from 33 °C on day 15, followed by a reversion to 31 °C, causing a male proportion of 55% of embryos in *A. mississippiensis*. On the same day, temperature changes ranging from 31 °C to 34 °C (followed by a return to 31 °C) led to a male population of 100% [18]. However, it is important to note that shifting tests may not always yield precise results due to variations in incubation durations and threshold temperatures among different species. Specifically, in *A. mississippiensis*, temperatures of 33 °C can result in a 100% male ratio, while low temperatures primarily lead to females [12], whereas other crocodile species such as *C. palustris, C. porosus*, and *Crocodylus johnstoni* do not display specific temperature patterns in their temperature-dependent sex determination (TSD) that result in a male-only ratio. This suggests a limitation in the TSD determination process in crocodiles [12]. Lang and Andrews (1994) concluded that the temperature-sensitive periods in crocodiles and alligators are quite similar but could vary in different species. This finding has been supported by subsequent studies on species such as *C. palustris, C. johnstoni, C. porosus,* and *C. niloticus* [12,20].

Not only the temperature but also the stages of the embryos influence the effectiveness of the temperature changes. Reptilian embryos are commonly staged based on their morphology. Ferguson (1985) standardized a staging system applicable to crocodilians, categorizing embryonic development into stages 1 to 28 [28]. Each stage has characteristic morphological changes. Based on that information, the determination of TSP in its coherence to gonadal differentiation can be applied [12,27]. For both *A. mississippiensis* and the spectacled caiman *(Caiman crocodilus)*, TSP was defined for the embryonic stages of 21 to 24, between days 30 and 45 during the middle third of embryonic development [12]. Joanen and Ferguson (1983) determined a smaller range for the TSP in *A. mississippiensis* rather earlier, between days 20 to 35 days of incubation [19]. A common finding, however, is that TSP in crocodilians has an approximate span of approximately 15 days [7,12,24]. The discrepancy in results between the two investigations could be attributed to the varied temperatures employed in the switching trials, as well as the modification of periods.

Temperature shifts before or later than the defined periods are deemed to be ineffective regarding sex ratios. Mostly affected by temperature shifts are sex ratios between stages 21 and 23 [4,12]. However, alterations at the beginning of the TSP appear to have the biggest effect on either inducing maleness or femaleness [4,7,20,23]. Incubation periods have varying lengths, notably controlled by the incubation temperature. Higher temperatures cause the length of incubation to decline, while lower temperatures extend the period [24]. Accordingly, the average incubation period of crocodilians varies between 70 and 80 days [21].

The TSP occurs during embryonic development and can therefore be linked to the stages of the incubation period. Temperature fluctuations occurring before or after the specified timeframe are considered to have no impact on sex ratios. The sex ratios between stages 21 and 24 are mostly influenced by temperature fluctuations, although this varies across different species [4,13].

### 2.3. Gonadal Differentiation

As mentioned above, TSP is the timeframe in embryonic development where the undifferentiated, bipotential gonads take a certain pathway into becoming either testes or ovaries, approximately between stages 21 and 24 [12,29]. Unlike in species expressing SD through GSD, the sex will ultimately be set during that time and not at the point of fertilization [10,30]. Post TSP, the sex of the embryo cannot be reversed [18]. The differentiation of gonads arises initially from the coelomic epithelium, or surface epithelium, which finally provides the basis for the “genital ridges”, by thickening of the epithelium [10,11,29]. Moreover, it forms the epithelial compartment within the gonad, holding the epithelial and primordial germ cells (PGCs). Those PGCs are primarily observable in the germinal epithelium and represent undifferentiated stem cells that can further develop into either oocytes or spermatozoa after traveling to the “genital ridges” [10,31]. While the genital ridges undergo development, epithelial cells from various structures proliferate. The coelomic and germinal epithelium gives rise to epithelial cells that align to form the inner part of the gonad—the medulla. Epithelial cells arising from the coelomic epithelium further take part in the formation of the anlagen for the rete cords. These cords are a primary connection between the embryonic mesonephros and the developing gonad [11]. Another key component in gonadal differentiation is the sex cords or the medullary cords [11,18,29]. These are formed by the proliferation of cells from the germinal epithelium and commonly occur during the early stages of TSP. The presence of sex cords is indifferent to the sex of the embryo [11].

When it comes to the effect of temperature on the primordial gonads, there appears to be a slight difference in female- or male-producing temperatures. While in lower female-producing temperatures, or high as in the case of *A. mississippiensis*, the medullary cords appear to be thinner and less infiltrated by cells, and the higher, male-initiating temperatures cause the cords to become thicker [11,18]. It is, however, not entirely possible at this state yet to determine whether the embryo will ultimately become male or female. One of the few indicators of a certain sex might be the pre-Sertoli cells, which were demonstrated mainly in *A. mississippiensis*. A higher amount seems to be a precondition for testes to develop [11,12]. Whether a gonad develops into an ovary or testes can be initially established by the state of the germinal epithelium. In the case of a male pathway, the epithelial compartment will show a flattened morphology, and germ cells will regress. In contrast, ovarian differentiation is highlighted by a thickened epithelium and proliferation of not only epithelial but also germ cells. Additionally, the formation of an ovarian cortex is highly characteristic, with a distinct separation from the medulla by a basement membrane [11,18,29]. Towards the end of TSP, most of the cells in a differentiated ovary will enter meiosis, while cells in determined testes differentiate distinctly as Sertoli cells by stage 23 [11,12].

### 2.4. Selective Advantage of TSD

It is still a topic of discussion whether the presence of a TSD mechanism is of a higher evolutionary advantage for reptiles, as opposed to the occurrence of GSD. The Charnov–Bull model, first described in 1977, suggests that reproductive fitness in the same environmental conditions is differently expressed in males vs. females and further highlights the significance of ESD evolution [22,32,33]. Bull and Charnov (1977) proposed that natural selection processes should benefit the hatchling for its most advantageous outcome [33]. These advantages can be related to weight, length, and most importantly maximizing reproductive fitness about sex. To better understand the Charnov–Bull model: it can be described as an imagined piece of land, with many different vegetative characteristics. On the far left of that land, you find a big river with surrounding pastures, extensively exposed to broad sunlight. On the contrary, the right side is mainly foresting territory, offering shadow and low-temperature spaces and lots of predator animals. These differences in the flora and fauna turn our fictional one piece of land into a lot of little islands—islands that can favor either the reproduction of female individuals or that of males [33]. Hence, the environmental conditions occurring in these little spots will favor the best outcome for the hatchling—in which GSD would not be possible, as it is not driven by the influence of the embryo’s surroundings [18,34,35]. Despite the difficulty of finding confounding evidence for the model, it poses the most realistic and applicable explanation as to why TSD has prevailed over GSD in certain species [10,33].

There are two ways to categorize the effects of incubation temperature on the embryo. On the one hand, it directly influences the outcome of the sex, and on the other hand, it affects the hatchling in various non-sexual relations. *C. niloticus* embryos that were incubated at warmer temperatures, at 34 °C, usually had a smaller body size compared to those incubated in cold and intermediate temperatures [20]. Hutton’s (1987) study was confirmed through similar observations made by Deeming and Ferguson (1989) on *A. mississippiensis* hatchlings. Offspring incubated at intermediate temperatures had the largest body size after hatching. As intermediate temperatures in crocodilians produce predominantly male progeny, the larger body size poses a selective advantage for the male sex. Larger-sized male crocodilians are usually at an advantage when it comes to territory fights against predators or achieving better reproductive fitness [18]. The increasing temperature stands in correlation with the accelerating growth rate of the embryos during incubation and post-hatching [18,21]. However, if the temperatures are close to the extreme limit, the growth and survival chances of the embryos decline significantly [18]. Such observations of post-hatching effects due to temperature act in favor of the proposed model by Bull and Charnov (1977) [9,33].

Despite size being an important factor regarding selective significance, body weight is less affected. One outstanding factor regarding body weight includes the egg yolk that forms throughout embryonic development. Significant amounts are detectable after the end of the TSP, at stage 25, where the yolk surrounds the embryo remarkably. Absorption of the yolk is a noteworthy factor in the development of the hatchling. It is predominantly absorbed into the abdominal cavity between stages 26 and 28, whereas stage 28 marks the time of hatching. How much of the yolk is absorbed and how it affects the hatchling are highly dependent on temperature [36]. Colder temperatures, which determine the female sex, typically produce heavier embryos, as they absorb more egg yolk. Less yolk is absorbed at higher temperatures. Additionally, growth rate might be positively correlated to the amount absorbed, as well as reaching sexual maturity [7,21,24]. Thus, a higher body weight due to a larger absorption of yolk is an advantageous precondition for female individuals to grow faster and improve their reproductive fitness. A higher number of sexually mature females, compared to that of males, creates an essential benefit in maintaining and expanding a population.

TSD allows for flexibility in adjusting sex ratios in response to environmental conditions. Over the lifespan of long-lived species (such as crocodilians), fluctuating environmental conditions (such as temperature changes) can lead to varying sex ratios in different cohorts, helping maintain a balanced overall population. It is because long-lived species have multiple reproductive cycles over their lifetime. This allows them to adjust to environmental changes across generations, ensuring their survival even in the face of drastic changes like climate warming. Their long lifespans provide a buffer period during which natural selection can favor individuals better suited to new conditions. In contrast, species with shorter lifespans benefit from GSD because it provides a stable and predictable sex ratio that is not influenced by external environmental factors. This stability is crucial for short-lived species, as they do not have the luxury of multiple reproductive cycles to balance sex ratios over time. Short-lived species can evolve more quickly in response to selective pressures because they have more generations over a given time. GSD allows for consistent sex ratios, which can be advantageous for rapid population growth and recovery. TSD has advantages in environments where resources are limited, producing more of one sex that has a higher reproductive value under those conditions. Climate warming, theoretically, poses a challenge for species with TSD because consistently higher temperatures can skew sex ratios towards one sex, potentially leading to population imbalances. However, the long lifespans of these species can help mitigate this impact over time as they can experience a range of temperatures across different reproductive cycles. However, individuals with short lifespans will benefit more from a genetically driven SD system, as they cannot adapt to changes over a long period [4,8,9].

In conclusion, the results on the significance of TSD and the correlation between the two major SD mechanisms are still missing, and no definite answer can yet be elicited to either of the questions. The Charnov–Bull Model (1977) provides the most substantial explanation as to why TSD has been established in the reptilian order and how it favors individuals that lack heteromorphic sex chromosomes [33].

### 2.5. Mechanisms of Temperature-Dependent Sex Determination and the Influence of Genes, Hormones, and Epigenetic Factors

TSD is a very complex system, composed of many factors, eventually leading to the differentiation of an undifferentiated gonad into either testes or ovaries. However, in GSD, there are obvious and exclusive components that we can associate with SD, and in TSD, these remain elusive [37]. The difficulty lies in finding a factor within the embryo that is sensitive to outer temperatures and reacts differently to either high or low climatic conditions. The stimulation of such factor would then eventually lead to a male or female outcome, by either suppressing or activating a certain cascade of molecular events.

In the next sections, we aim to provide a comprehensive explanation of the complex and difficult process of activating environmentally influenced genes, generating hormones, and modulating epigenetics. These components have been previously identified as important in determining sex (Figure 2).

#### 2.5.1. Genes and Proteins Involved in Temperature-Dependent Sex Determination

The most recent investigations proposed a hypothesis that suggests the cellular sensor responsible for initiating ESD is initiated by a balance between redox regulation and calcium (Ca^2+^) signaling. This theory suggests a connection between calcium-sensitive cellular signaling and epigenetic changes that have been associated with temperature sex reversal. Although this theory was first observed and described in turtles [38], many members of this pathway were identified in crocodiles.

The hypothesis suggests that, in ESD systems, an alteration in intracellular Ca^2+^ levels (likely facilitated by thermosensitive transient receptor potential TRP channels) and an elevation in reactive oxygen species (ROS) levels due to high temperatures modify the state of the cell. This modification then initiates cellular signaling cascades that lead to the differential expression of sex-specific genes, ultimately determining the sex of the organism. These genes might be the first participants in the determination of crocodilian sex determination [39]. In mammals, there are well-known transcription factor genes involved in the process of SD. The most important factors include SRY, the steroidogenic factor 1 (SF-1), and Sry-related HMG box gene 9 (SOX9). The SRY gene serves as the key activator molecule in determining the sex of an individual in GSD. These factors also show temperature-specific expression patterns in slider gonads during TSP [40].

TSD is generally a complex procedure, involving multiple different steps and responses to environmental influences. Through cloning and isolation, it has been established that, in alligator embryos, several analogs to those mammalian SD genes can be found, but with the key factor, the SRY, or in mice Sry, which is located on the Y chromosome, no homolog was found in crocodilians [1,4,41,42].

##### The Temperature-Dependent Calcium Channels: The Transient Receptor Potential Family

Contemporary understanding of the molecular approaches for triggering genes and molecules in response to temperature fluctuations have focused on the transient receptor potential (TRP) family. This gene family of ion channels has gathered considerable attention in the study of thermosensory discrimination mechanisms (TSDs), owing to its high susceptibility to heat stimuli and the resulting elevation in calcium (Ca^2+^) levels [43,44]. Five of those thermosensitive ion channels, called transient receptor potential vanilloid type (TRPV) channels, were identified in *A. mississippiensis* by Yatsu et al. (2015) [43]. Out of TRPV2, TRPV4, TRPM3, TRPA1, and TRPM8, the TRPV4 channel proved to have the most temperature sensitivity in *A. mississippiensis* and *C. moreletti* [43,45]. In the female gonad, TRVP4 is commonly expressed within the steroidogenic follicular cells. In the testes, Sertoli cells and localized cells around the testicular cords are responsible for the synthesis of the protein [45]. Since the TRPV4 gene and the channel are argued to be a possible thermosensitive triggering factor in TSD, it is of interest to determine how exactly it occurs during embryonic development and if major differences are occurring between male- and female-differentiating gonads.

Concerning the male-determining pathway, TRPV members have a suspected up-regulatory function for SOX9 in crocodilians. Evidence, based on TRPV4 antagonist injections into *A. mississippiensis* embryos at stage 19, suggests that the downregulation of the ion channel negatively affects the expression of SOX9 and Anti-Müllerian hormone (AMH), with both genes majorly involved in testicular differentiation. The application of agonists into a bipotential gonad did not result in significant alterations, only a mild elevation in the SOX9 gene expression. This suggests, again, that there is a direct connection between the heat-sensitive ion channel and genes involved in the testis differentiation, but also this indicates that female differentiation is minorly or not at all affected [43,45].

Upon thermal stimuli, TRP channels commonly cause an increased intracellular influx of Ca^2+^ [44]. This specific reaction might be an interesting relation with signal transduction molecules, like the Signal Transducer and Activator of Transcription 3, short STAT3 [44], because, in turtles, STAT3 acts as an inhibiting factor for an epigenetic regulator, which is considered to be in connection with male determination. STAT3 itself is activated by phosphorylation at warmer, female-incubating temperatures through the influx of Ca^2+^ [46,47]. However, in *Alligator sinensis*, the gene TRPV4 does not exhibit a biassed expression based on sex during TSP. On the other hand, three additional genes from the TRP family, namely, TRPV2, TRPC6, and TRPM6, show varying levels of expression in the male or female gonads during mid-TSP [48].

The correlation between the Ca^2+^ influx and the consequential upregulation of a sex-determining factor raises the attention provided to TRPV channels, which indeed causes an intracellular increase in Ca^2+^ in response to higher temperatures and induces signal transduction, like STAT3 in turtles [43,45,47].

##### Signal Transduction Molecules: JARID2 and KDM6B

In addition to the signal transduction, there are other mechanisms influenced by temperature that can modify gene expression in crocodiles, after initiating gene expression. Bock et al. [49] in their 2020 study measured the extent of temperature changes that alligator embryos experience in their natural nests and conducted experiments to observe how the expression and alternative splicing of genes related to sex determination and sexual differentiation change during daily temperature fluctuations in TSD. This investigation revealed that the process of alternative splicing of the epigenetic modifier JARID2, occurring in both the gonad and brain, exhibited prompt responsiveness to changes in temperature. In addition, the levels of both JARID2 and KDM6B gene transcripts in the gonad varied with temperature and were considered to play a significant role in the TSD due to gene expression influences [49].

##### Transcription Factors as Regulatory Molecules: The Role of SOX9 and DMRT1

In addition to the gene regulators mentioned, several transcription factor genes were found to be responsible for sex determination and differentiation in crocodiles, namely, the SOX9 gene. These transcription factors are related and highly conserved genes in vertebrata. In mammalian vertebrates, a specific cascade of genes with specific regulation patterns are described as sex-determination factors. The SOX9 is upregulated by SRY and further influenced by SF-1, a testis-differentiation-specific gene, which acts as an upstream regulator of AMH, one of the male determination factors, by positively affecting the expression through Sertoli cells in the gonad. SF-1 and SOX9 can be viewed as co-workers in the upregulation of AMH [11,50,51,52]. In crocodilians, instead of SRY, SOX9 was successfully isolated and cloned in reptiles, including crocodiles and alligators, while SRY was found to be absent. Due to its highly conserved nature, the organization of alligator SOX9 and that of *C. palustris* shows very similar patterns for SOX9 of birds and humans, and therefore, comparable purposes can be demonstrated [42,53,54].

Regarding the regulation pattern, SOX9 and DMRT1 gene levels themselves did not show distinct levels at different temperatures. Indeed, the temperature influenced their gene products, as evidenced by the observation that the SOX9 gene was found to go under alternative splicing, leading to distinct post-transcriptional products. Different protein levels were observed in male and female progeny, which can be related to distinct post-transcriptional variations [53]. These findings indicate that varying temperature signals encountered in the natural environment are combined through a sequential series of chemical reactions during temperature-dependent sex determination (TSD) [49].

In mammals, male sex determination and testis differentiation are highly dependent on Sertoli cells that usually are present first in an immature state and then further differentiate into mature and fully functional Sertoli cells [11,12,42]. Just like AMH, SOX9 expression was detected in immature pre-Sertoli cells and the differentiated mature ones. The specific location of expression travels from the cytoplasm into the nucleus of the cells [45,50,51].

In embryos of *A. mississippiensis,* Western and Graves (1999) mapped the activity of SOX9 and AMH through the timeframe of the TSP, from stage 21 to stage 24, both at male- and female-producing temperatures [42]. According to the presented results, right before the onset of TSP, the gonad is undifferentiated and has the potential to commit to either the male- or female-differentiated pathway [42,51]. AMH was already expressed by pre-Sertoli cells at stage 22, as opposed to SOX9, which was detectable at stage 23. Neither of them was identifiable at high nor low female-producing temperatures. Both genes peaked in their expression at stages right after the TSP came to an end and were primarily observed in fully differentiated Sertoli cells, which were finely organized into testis cords [42]. SOX9 being a testis-specific factor is further highlighted by the application of exogenous E2 into eggs incubated at male-promoting temperatures. In response to the steroid hormone injection, SOX9 and AMH are significantly downregulated, while ovarian stimulants like aromatase (AT) are noticeably increased [41]. Overall, SOX9 in *A. mississippiensis* does not seem to have any upregulating function on AMH; therefore, it does not passively initiate the regression of Müllerian ducts and consequently is possibly not the crucial factor in male determination. If anything, SOX9 might be viewed as a differentiating factor, but not a key determining one, concerning TSD [42,54].

##### Nuclear Receptors as Differentiation Markers: SF-1

Apart from SOX9, there is one more major regulatory gene, which is also involved in the mammalian SD process. SF-1 belongs to the family of orphan nuclear receptors and relates to the embryogenesis of mammals and non-mammalian vertebrates. It is present in the genital ridge of mammalian embryos and expresses a sexually dimorphic pattern, meaning it is present in both male and female gonads [11,55].

The primary function of SF-1 within SD can be viewed from the male-determining or the female-determining pathway, as activity is detectable in ovaries and testes. Generally, SF-1 is produced in supporting cell lineages, but their expression is mainly detectable in Leydig cells of the differentiating testis [50]. The role of the gene is primarily limited to the direct regulation of proteins included in steroidogenesis and the upregulation of AMH along with SOX9, by acting on the Sertoli cells [11,42,55,56].

In what way exactly SF-1 belongs to the SD mechanism in crocodilians was demonstrated by Western and Graves (2000). Although the authors focused primarily on the role of SF-1 in the gonadogenesis in embryos of *A. mississippiensis*, there is a strong indication to assume that the expression and function of SF-1 in crocodiles would deliver similar results, as the gene has been strongly maintained and conserved amongst vertebrates. Adding to that, cloning of SF-1 homologs in *A. mississippiensis* revealed a strong similarity towards the human orthologue [57]. During the early stages of embryonic development, where the gonad is yet to differentiate, SF-1 shows a clear sexual dimorphism and is detectable around stage 20. The onset of TSP in *A. mississippiensis* is set to stage 21 but is not an exclusively male- or female-differentiating factor, and thus, it is most likely involved in feminizing and masculinizing processes [57]. SF-1 is noticeably downregulated in the embryonic gonad at male-producing temperatures during the temperature-sensitive, critical period of *A. mississippiensis*. The low concentration of SF-1 in the beginning stages of gonadal differentiation is quite similar to the expressional pattern of SOX9. Both SOX9 and SF-1 are thought to have regulatory functions on AMH [51,58]. The upregulation of AMH before SOX9 and SF-1 in *A. mississippiensis* raises the question of the regulatory relationship between the genes; however, low amounts of SF-1 might be sufficient to significantly upregulate AMH [57,59]. Additionally, SF-1 positively influences the synthesis of androgens, which are essential in the male-differentiating pathway [55,56].

However, none of these findings definitively concluded that SF-1 is exclusively responsible for determining either the testis or ovary and that it is solely associated with one sex, like in mammals [11,55]. Although there may be higher levels of male or female gene expression, this does not diminish the existence of mRNA in the opposite gonad.

#### 2.5.2. Hormonal Influence in Temperature-Dependent Sex Determination: Steroid Hormones and Aromatase

The previous findings suggest that hereditary variables alone are insufficient to fully explain TSD, while other factors, such as hormones, also have a substantial role in affecting these occurrences. During the development of the ovary in embryos, genes involved in the production of steroid hormones are expressed at a high level. This suggests that steroid hormones play a crucial role in determining the female sex [48].

Multiple articles have provided comprehensive information regarding the existence and impacts of hormones.

##### Steroid Hormones

The first hypothesis regarding the molecular mechanism behind TSD includes the involvement of steroid hormones, specifically 17-beta-estradiol (E2) and testosterone (T). Even though numerous pieces of evidence have confirmed that steroid hormones are involved in gonadal differentiation, their exclusive role has not yet entirely been put into perspective [10,60]. As an initial theory, Janzen and Paukstis (1991) proposed that their association might lie in the ratio of E2 to T, ultimately deciding the fate of the gonad to become either an ovary or testis. Considering that, the application of exogenous steroids, such as E2 and T, was effective in altering the ovarian differentiation at female-producing temperatures or overriding the male-producing temperatures in *A. mississippiensis* [4,10,17] and *C. porosus* [8,61,62].

In addition, the investigations conducted on the common snapping turtle, *Chelydra serpentina* [11], showed that injections of E2 and E2 agonists at temperatures that promote male characteristics triggered the development of ovaries in the gonads. However, the use of T and dihydrotestosterone (DHT), a steroid that cannot be converted into estrogen [11], at female incubation temperatures gave mixed and inconclusive results, in comparison to the E2 injection at male-promoting temperatures. Masculinization through injection of DHT is therefore questionable [63,64,65]. In alligator embryos, injection of E2 antagonists, such as Tamoxifen, gave contradicting results by occasionally acting as an E2 agonist, but did not interfere with the ovarian development [31,60].

Endogenously, steroids are provided by the egg yolk during embryonic development. In *A. mississippiensis*, E2, T, and androstenedione (A), which can be converted to either E2 or T, are present even before the TSP begins, with A having the highest concentration. Higher concentrations of A at the beginning of the TSP are coherent with its conversion later during the gonadal differentiation. Between stages 21 and 23, the overall number of steroids declines, independent of the incubation temperature [60,64].

##### The Role of Aromatase

The ratio of sex steroids is associated with the activity of the CYP450 enzyme aromatase (AT), which is encoded by the CYP19 gene. The primary role of AT is the conversion of T to E2 and A to estrone (E1) [11]. It is assumed that the activity and/or synthesis of AT is correlated with the incubation temperature during the TSP. The following is a simple way of explaining the role of AT in SD: if the ratio of estrogenic to androgenic steroids is high, it determines the fate of the female pathway and vice versa [8,29,59,66]. Injections of AT inhibitors (AI), e.g., Fadrozole, in eggs of *Emys orbicularis* turtle achieved the inhibition of ovarian development at female-producing temperatures. Furthermore, successful masculinization, characterized by a thickened germinal epithelium rich in Sertoli cells [30], was demonstrated in *C. porosus* and *A. mississippiensis* by using 4-Hydroxyandrostenedione (4-OHA), by markedly reducing the activity of AT [62,64].

Concentrations of AT are commonly measured within the gonad–adrenal–mesonephros (GAM) complex and in the brain tissue of reptilians [67]. *A. mississippiensis* embryos at stages in the beginning and during TSP only show a low activity of AT in their gonads, independent of the incubation temperature. Significant increases are primarily recognizable towards the very end of the TSP, and most typically at stage 24, post-TSP [39,67,68]. Similar results are observed in *C. porosus*, where AT reaches its peak concentration at stage 23, when the TSP has come to an end [61]. Regarding brain tissues, when the TSP concentration of AT is higher compared to that in the undifferentiated gonad, there is no notable difference between male- and female-producing temperatures. Based on that data, it is doubtful whether AT activity in the brain is a pivotal factor in the mechanism of TSD [39,67,69].

The gonadal concentrations of AT during the TSP indicate that the enzyme might not ultimately be the determining factor for a male or female pathway. AT increases only after TSP at female-promoting temperatures in *A. mississippiensis* [39,67]. The proposed architecture of AT activity during TSP suggests that temperature does not directly impact the enzyme’s activity. Instead, it influences the synthesis of the enzyme by affecting higher stages within the SD cascade. A more logical conclusion is that increasing levels of AT are not a reason for gonadal differentiation, but rather a consequence of it [68]. However, the upstream event leading to the increase is still subject to discussion. Joanen and Ferguson (1983) proposed in their study on TSP that, for sex to be determined, there should be a male-determining factor that reacts sensitively to the external temperature and consequentially initiates a cascade that will lead to either the activation or inhibition of the male-inducing factor, resulting in female development. Since all crocodilians express the TSD2 pattern, the factor should therefore be active in intermediate temperatures and pose an inhibitory effect during high and low climatic conditions [39]. The fact that, in crocodilians, AT is present during embryonic development, but only rises after gonads have developed into ovaries seems to support that hypothesis [39,68].

##### Anti-Müllerian Hormone

AT is inhibited by the anti-Müllerian hormone (AMH), which is exclusively produced by Sertoli and Leydig cells in the testis [70]. The primary function of AMH is the regression of the Müllerian ducts, an essential part of the masculinization during gonadal development. Without the effect of AMH, the ducts develop into necessary parts of the female genital tract [50,57,71,72,73]. Aligning with the pattern presented in AT, AMH peaks in its concentration after TSP seems to be indetectable at stages of a bipotential gonad. First, detectable values occur from stage 22 onwards [57,58]. Considering that no AMH is detectable while the gonad is still in its undifferentiated bipotential state before TSP, it is undeniable that it is specific to the male sex-differentiation process in the gonads. Nonetheless, regardless of the effects exhibited by the hormone, it is possibly not the initiating factor for male determination. Western et al. (1999) examined the correlation of testis differentiation to the activity of AMH, with special regard to the gonadal development in *Alligator mississippiensis* embryos [42]. They detected that, even though AMH was active during the early stages of TSP, gonadal tissue arrangement towards a male pathway was initiated earlier, during stage 21. It is therefore possible to assume, just as with AT, that AMH itself is not at the forefront of male determination but is rather the consequence of a previously occurring event. Additionally, Western et al. (1999) failed to detect AMH at female-promoting temperatures, neither high nor low [42]. This suggests that the occurrence of AMH is fully limited to a testis-determining gonad and entirely relies on the presence of premature and mature Sertoli cells. Concerning the timing and concentrations of AMH, there is a strong suggestion that it in fact can override the effect of AT in the gonad, which is not only limited to reptilians. Similar results were obtained in ovine embryos injected with bovine AMH. The activity of AT in the gonad was fully inhibited [74]. Apart from AMH, a possible regulatory origin of AT lies in genes and environmentally influenced signals involved in SD, which either suppress or activate the responsible gene of the enzyme [40,72,75].

#### 2.5.3. Epigenetic Factors and miRNAs

As mentioned before, the TSD of crocodilian species is influenced by various parallel events. The diverse phenotypes can be attributed to the interplay of genes, hormones, and epigenetic modifications. These factors dynamically respond to environmental stress processes, influencing the expression and function of genes and hormones. Based on the variability in sex development between distinct crocodile species, which does not exactly comply with previously established patterns, the researchers have discovered multiple epigenetic modifications that affect the previously mentioned key factors.

In a genome-wide methylation experiment, an association was found between incubation temperatures and differentially methylated loci. The researchers investigated differentially methylated cytosines (DMCs), which are highly methylated, and found a temperature-associated pattern [49].

Parrott et al. [76] discovered that the methylation patterns in the CYP19A1 promoter region of the SOX9 gene demonstrate sexual dimorphism and suggested that the methylation occurs due to temperature changes. They also observed sexually dimorphic epialleles, which revealed a new differentially methylated region (DMR) upstream in an organism that undergoes temperature-dependent sex determination [76]. The promoter region is predicted to have a regulation role in aromatase production, but it was only described in turtles [40]. The data reported indicate that DNA methylation patterns play a role in connecting incubation temperature to a sex-specific genetic program, which determines whether an individual produces a testis or an ovary. A dimorphic expression of the SOX9 gene and the AMH could be measured in the embryonic stages 21–24 in American alligators [77], so the methylation was assumed to happen before this period. The AMH and the aromatase from the hormonal determination factors do not have any CpG islands on their promoters, so the regulation of these components could not happen due to methylation [76].

Besides the methylation of the DNA, microRNAs play a role in the regulation of sex determination. Two miRNAs from the miR-10 family, specifically miR-10a and miR-10b, exhibited a higher expression in females during mid-TSP. These miRNAs targeted two genes involved with male sex differentiation, ADCY4, and FGFR2, which are a part of the signal transduction pathway that activates gene proliferation. Post-transcriptional sexual dimorphism was seen in the expression of two members (miR-133a/b) of the miR-133 family. These two microRNAs target many genes associated with sex determination, specifically those involved in the processing of steroid hormones [48].

## 3. The Role of Climate Change

The earliest traces of crocodilian existence by the analysis of fossils can be traced back to the Late-Cretaceous period, around 95 million years ago [78,79]. With the catastrophic cretaceous-tertiary event (K-T event), a major cleft was left in the ecosystem across the globe, leaving animal species the choice to either adapt or to inevitably face extinction. Despite the closely associated relationship between crocodilians and dinosaurs, both belonging to the archosaurs, it is curious that only one of the two managed to prevail into modern centuries. Additionally, the phylogenic similarities leave one to wonder, whether dinosaurs also used TSD and were, however, not capable of adjusting their adaptiveness to a new world, post the K-T event [17,78,80].

One way or another, history proved that crocodilians could adapt their population to significant changes in their environment. With global warming and climate change becoming a more significant issue in recent times, there is no denial in the fact that species using TSD are at a high risk of facing extinction. According to the World Meteorological Organization, global surface climate will be around 1.1 °C to 1.8 °C higher when compared to the detected temperatures in the 1900s [81]. This is without considering the effect of human-produced greenhouse gases. Natural catastrophes caused by climate change are detrimental to our ecosystem. Flooding and bushfires cause unthinkable damage to the vegetation and natural habitats of animal species. Rising sea levels additionally force land species to expand to alternative geographical locations.

Taking all of this into consideration, how is SD a crucial factor in the face of climate change? Species that determine their sex via chromosomes are at no risk of their sex ratios within a population being altered by environmental stimuli, such as increasing temperature, or other influencing factors. The sex of the offspring is set at fertilization and cannot be reversed or overridden [78,82]. TSD species, on the other hand, depend on the outer temperature during egg incubation. Any changes in PT or TRT of even 1 °C could drastically change the sex ratio of the population. A tendency towards warmer climates usually results in a ratio that is biased towards more females, or possibly also an overproduction of males, if temperature shifts fall within the male-producing temperatures. Such predictions were made for the population of *A. mississippiensis*.

Bock et al. (2020) [49] forecast a temperature increase in nests within the geographical contribution, along the East Coast of America, around 1.1 °C to 1.4 °C by 2050. Mean temperatures are currently 32 °C, favoring the production of male offspring. As *A. mississippiensis* is known to use the TSD2 pattern, increases in such temperatures would initially not drastically change the principle of progeny. Males would still be produced, as they commonly still hatch at 100% around 33 °C. However, due to the occurrence of less low-temperature females, the male/female ratio is expected to bias significantly towards males, around 90% [49]. The lack of females within the population could pose a significant problem, as a high number of reproductive cycles of the few female individuals would be needed to restore a balanced sex ratio [49,78,82].

Considering that global warming is not a stagnant process, and under the precondition that the alligator population would be able to overcome such initial skew in ratios, climatic conditions and their effects on the population will become even more drastic. A further prognostic by Bock et al. (2020) states that, by 2100, nest temperature will be 4.6 °C higher compared to today. With this being true, males at intermediate temperatures would only seldom be produced and the sex ratio of *A. mississippiensis* would shift drastically, from 90% males to not more than 2% [5,49].

Due to this reason, realistically, crocodilians are at the edge of extinction; however, their extinction is hard to forecast. As of right now, eight crocodilian species are regarded as critically endangered, as listed on the red list by the International Union for Conservation of Nature (IUCN). Further, climate change and the consequential relocation to different areas are listed as an explicit reason by the IUCN for the categorization as critically endangered, apart from hunting and invasion of wildlife by humans. Although their SD mechanism lacks certain variety, due to the absence of sex chromosomes, knowing that crocodilians already survived a global catastrophe once, there is hope and confidence in assuming that they could do it again, even by using TSD as an SD mechanism [49,78,82].

A possible way for them to adapt to the change that our globe is facing is by the simple tool of geographical migration. As the Bull and Charnov (1977) model suggested, the primary evolutionary advantage of TSD lies in the ability of animals to choose which patches in our environment are most favorable for their outcome. If natural habitats are destroyed by bushfires or floods, those original suitable patches might not be available anymore; hence, crocodilians need to find new areas to live in. Such relocation and migration have been documented by fossil records over crocodilian history, showing that they can adapt to changing circumstances [78,83]. Currently, several natural habitats by crocodilians are affected and critically put at risk. *A. mississippiensis* has its primary geographical contribution located from Northern America down to the East Coast and is a common inhabitant of the Everglades in Florida [49,84]. A suspected increase in temperature of around 2 °C to 2.5 °C could significantly alter the flora and fauna of the wetland of the Everglades. Droughts would leave *A. mississippiensis* with fewer available nesting sites and reduced food availability, resulting in a decreased reproductive function [84].

Besides the temperature increase, Fukuda et al. (2022) examined the effect of rising water levels in *C. porosus* along the northern coast of Australia. The crocodile species mainly reside in freshwaters, like rivers and lakes; thus, a mixture of water quality due to seawater invasion into freshwater habitats could result in a reduction in possible living areas [85]. The authors conclude that sea levels in that area are possibly going to rise around 1.90 m by 2100, resulting in ca. 60% less available natural territory for *C. porosus*.

Overall, climate change is a real and menacing risk for the crocodilian species around the world. Still, due to their resilient nature and the new knowledge about TSD, which was gathered over the last few years of research, there is a good chance of being able to preserve these large reptiles Although crocodilians primarily depend on TSD, the mechanisms by which temperature influences sex are obviously encoded in their DNA. This does not mean they have an alternative GSD system that can override TSD if environmental conditions fail to produce balanced sex ratios. The genetic basis of TSD involves specific genes responding to incubation temperatures, but crocodilians lack a secondary genetic system like GSD to directly determine sex. However, PT and TRT work in the reptiles’ favor, by still producing many males at high temperatures and ultimately females at even higher temperatures (please refer to Figure 1). Additionally, the long lifespan and the possibility of having many reproductive cycles to reinstate a balanced sex ratio are significant advantages and should not be underestimated [4,78,82].

## 4. Conclusions

SD mechanisms in our ecosystem have proven to be very different and are utilized quite diversely among the animals around the globe. By analyzing the results that were collected over the past few years, it becomes apparent that crocodilians represent a special clade of reptiles. Their offspring’s sex is determined by the temperature during their incubation time. Females are commonly produced at high and low temperatures, while males predominantly hatch when temperatures fall into intermediate levels. Variations between species exist, but generally, the patterns apply to all members of the crocodilians. Researchers have proven that the sex of eggs is not only determined by temperature in artificial incubation experiments but that it is also a common event in nature. Therefore, clutches had different sex ratios, all depending on the location of the egg within the nest and the environmental temperature.

Even though TSD has been the focus of research ever since Charnier (1966) discovered the odd mechanism in a lizard, no clear explanations have yet been given on how exactly temperature poses its effect during embryonic development. Multiple hypotheses have been proposed, with surely all of them being crucial components in the differentiation of the gonad during the TSP. Steroid hormones such as E2 and T are in direct correlation with sex, as their expressional patterns align with either a female or a male hatchling. Furthermore, AT, an enzyme that knowingly converts T into E2, shows higher activity levels in putative ovaries. Adding to that, the application of AI causes the interruption of ovarian differentiation, even at female-producing temperatures. Genes involved in mammalian SD were also isolated and determined in crocodilians and turtles. Their expression varies slightly between the species, but their role in TSD during the TSP is undeniable. However, what is missing is the pivotal thermosensitive factor that ultimately elicits a response in the bipotential gonad that will either cause the male- or female-determination pathway to be induced. Thus, none of the proposed factors, neither steroid hormones nor genes, are solely responsible for the determination of the gonad into the testis or ovary, indicating that they must be regulated by other effects such as epigenetic factor or post-transcriptional modification.

The significance of TSD is highlighted by its post-hatching effects on the hatchlings. Females with higher body weight, due to yolk absorption at lower temperatures, benefit from reaching their reproductive fitness quicker than males, hence having many reproductive cycles to balance the population. On the other hand, adult males benefit from a large body size, by having an advantage because of their territorial behavior.

Overall, there lies significant importance in understanding how TSD operates among species. Being able to determine the TSP and the critical temperature ranges is essential in the possible preservation of endangered species that face the danger of climate change. Even though such drastic effects seem to be in the far future, there is a realistic chance for animal species to be critically affected by rising sea levels and continuously increasing global temperatures. Further research in the field of TSD is consequently indispensable, especially since the major thermosensitive factor in the SD mechanisms remains elusive.

## Figures and Tables

**Figure 1 animals-14-02015-f001:**
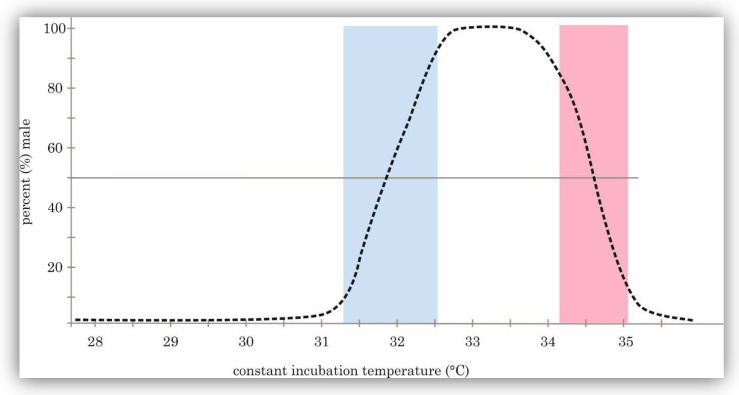
TSD2 pattern of *A. mississippiensis*: Males have a low proportion at decreased and very high temperatures, while at the intermediate temperatures, higher male ratios can be observed (dotted line). The blue and rosa bands describe the development of the PTs where a 50% male proportion was observed at 31.7 °C and 35.9 °C, while at 33.4 °C, the highest male ratio was found, according to the data of González (2019) [13].

**Figure 2 animals-14-02015-f002:**
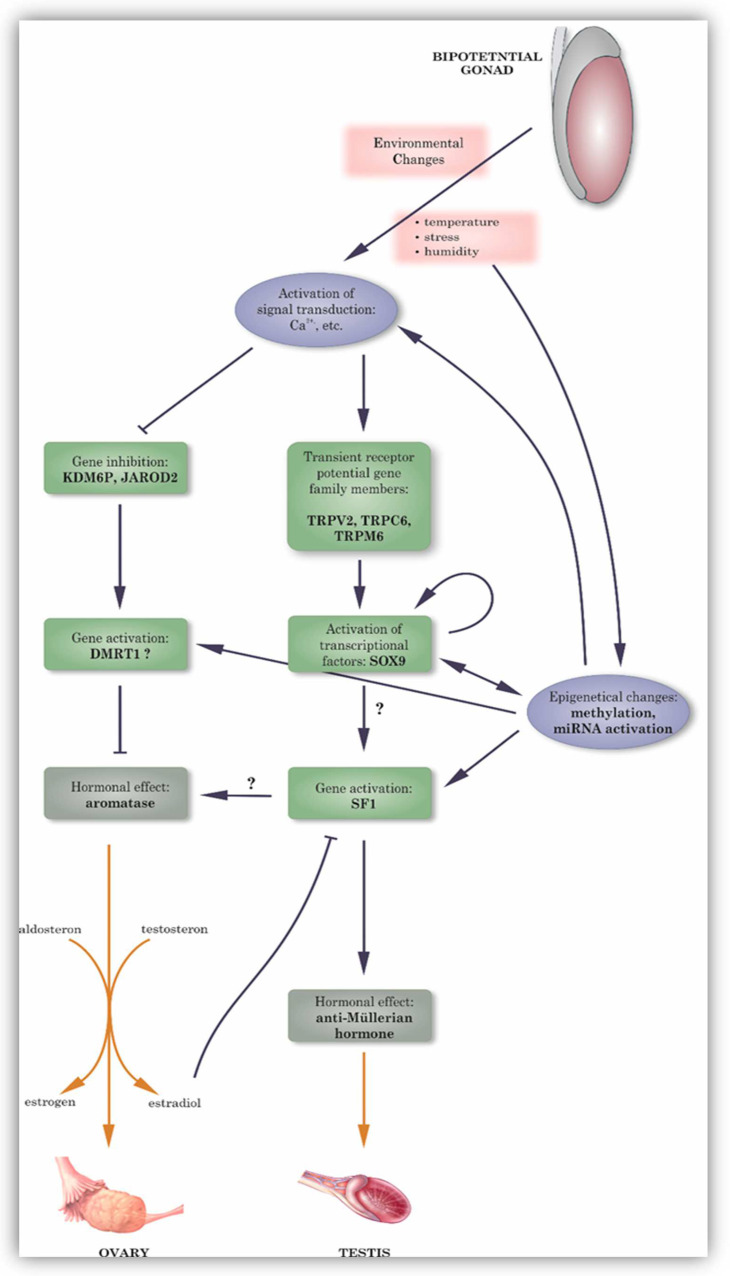
Schematic figure for summarizing the involvement of steroid hormones, enzymes, and genes in the TSD mechanisms.

## Data Availability

No new data were created or analyzed in this study. Data sharing is not applicable to this article.

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
