# Peer review of "Temperature-Dependent Sex Determination in Crocodilians and Climate Challenges"

_animals, 2024, doi:10.3390/ani14132015_

Round 1
Reviewer 1 Report
Comments and Suggestions for Authors
The paper on Temperature-dependent sex determination in crocodilians and climate challenges is a very ibìnteresting review providing a useful updating on infkuenxe of climaric cgane on sexdeterminatio in crocodiles.which can be accepted fro publicatio in its present form.
Author Response
Dear Sir/ Madam,
thank you for your review.
Best regards
Reviewer 2 Report
Comments and Suggestions for Authors
Initially, I would like to extend my compliments to the authors for their remarkable efforts in the field of reproduction in crocodilians, as well as for their dedication in preparing this comprehensive review on sex determination in crocodilians. The comments and criticisms presented below were made with the best intentions, aiming to contribute to the enhancement of this manuscript.
Overall, the text is very well written. However, there are some errors in the way other studies are cited and duplicated references. These items will be addressed at the end of this reviewer's comments.
From this reviewer's perspective, the study lacks a practical application of the knowledge synthesized in this review. The IUCN works with the One Plan Approach, integrating in situ and ex situ actions. Recently, a more comprehensive concept has been developed: "One Conservation." This concept aims to highlight the importance of changing the paradigms of other parties that have never been involved in conservation programs but are fundamental to ensuring not only the conservation of species but also of ecosystems. Defined as "an interconnection between ex situ and in situ conservation plans, anthropic actions on the environment (sustainability), and research in different areas that encompass conservation," this interconnection is discussed in greater depth in the integrated vision part of the article, where we point out the current gap between agribusiness and the conservation community. https://doi.org/10.1590/1984-3143-AR2021-0024.
This reviewer recommends that the authors, based on the One Plan Approach or, even better, on the more comprehensive "One Conservation" model, propose how to direct, in ex situ environments, the proportion of births towards one sex or the other, which could be beneficial to compensate for in situ populations impacted by climate change, by reintroducing individuals born in ex situ environments into the wild. In other words, how the integration of conservation actions (including commercial crocodilian farming) could compensate for—or better, minimize—the impact of climate change on the conservation of crocodilian populations. This is the main point suggested by the reviewer for the manuscript, aiming to present not only the problem but also mitigation strategies. This could enrich the manuscript.
Specific Points:
Line 48-50: It seems that a reference is missing for this statement (considering it is a sentence in the Introduction section).
Line 82 and also Line 113-114: The reference number [14] is missing after Gonz-alez (2019). By the way, it is Gonzales et al. (2019).
Line 111: Ferguson (1983) is missing the reference number [xx]. If it is reference [28], it should be Ferguson & Joanen (1983) and not Ferguson (1983) and colleagues.
Line 185-187: The sentence seems out of place. Please review.
Line 196: What does "pulses" mean?
Line 198: I understand that "antipredator behavior" occurs after birth. This sentence is difficult to interpret. It is not easy to understand what the authors are conveying. Also, what is the reference for this information? It is neither in reference [13] nor in [32].
Line 199-200: Do smaller males have fewer chances of reproduction? I did not find this information in the cited reference [32].
Line 316-317: Eggs incubated at 34°C produced smaller hatchlings compared to lower temperatures. However, the incubation time at 34°C was shorter, correct? Therefore, the incubation time could not be the factor affecting hatchling size, as mentioned in lines 324 and 325.
Line 323: Is Alligator mississippiensis not an apex predator in its in situ environment? Besides humans, what would be the predators? I believe that Ursus americanus and Puma concolor could be potential predators of A. mississippiensis; however, both species are practically nonexistent in the range of A. mississippiensis. Additionally, U. americanus would prey on eggs and, at most, very young animals. P. concolor is not a species with much affinity for water, and even if predation documentation exists, it would be small and young animals that are more vulnerable.
Line 350-351: It is great that this sentence was included. It was the impression this reviewer had after reading this section!
Line 368-370: The schematic figure is very interesting!
Line 684-685: "Species that determine their sex via chromosomes are at no risk of their sex ratios within a population being altered by environmental stimuli, such as increasing temperature." Although it is true that genotypic sex determination is not directly affected by temperature, extreme environmental factors can still impact the survival and reproduction of these species, indirectly affecting sex ratios. For example, exposure to environmental pollutants such as endocrine disruptors can alter sex ratios in some fish species, even if the sex is genetically determined. Extreme environmental factors can impact sex ratios in species with genotypic sex determination through mechanisms such as differential mortality, differential reproduction, and interference in embryonic development. These impacts can result in a disproportionate sex ratio, affecting the population dynamics and long-term survival of these species. In summary, the statement in the context of the article is correct but very strong. This reviewer suggests that the authors consider restructuring the sentence.
Line 700-702: "The lack of females within the population could pose a significant problem, as a high number of reproductive cycles of the few female individuals would be needed to restore a balanced sex ratio." This statement is correct, but it should be noted that the ability of a population to recover from a sex ratio imbalance depends on many factors, including the survival rate of the offspring and the ability of females to reproduce successfully.
Line 744-745: "Crocodilians might not have a possibly genotypic SD in their background, which could take overriding action if TSD fails." This statement can be a bit confusing. Although crocodilians primarily depend on TSD, this does not mean they lack genetic mechanisms that influence sex determination. In lines 348 and 349, it is stated that "Individuals with short lifespans however will benefit more from a genetic driven SD system, as they can’t adapt to changes over a long period." It would be important to review this information.
Errors in Citations Throughout the Text:
Missing Reference Number (and, in some cases, the Year):
Line 437: Bock et al. [58] was used. The reference should replace the year, which seems to be the correct format for this journal.
Line 161
Line 182-183
Line 192
Line 195
Line 220
Line 226
Line 294: Missing reference for the Charnov-Bull model.
Line 297
Line 317
Line 319
Line 352
Line 474
Line 504
Line 542
Line 594
Line 762
Errors in Duplicate References:
Reference 4 is the same as Reference 45
Reference 9 is the same as Reference 18
Reference 13 is the same as Reference 33
Reference 16 is the same as Reference 22
Reference 20 is the same as Reference 29
Reference 23 is the same as Reference 36
Reference 24 is the same as Reference 37
Reference 26 is the same as Reference 27
Reference 28 is the same as Reference 35
Specific Reference Correction:
Reference [93]: Change the URL to DOI, as it is an article and not a webpage.
Author Response
Dear Sir/Madam,
thank you for your review. Please see attached our respond to your comments and suggestions.
Kind regards

Reviewer 3 Report
Comments and Suggestions for Authors
The author conducted a literature search using keywords to review temperature-dependent sex determination (TSD) in crocodilians. However, based on my experience and other review articles on TSD I have read, the language logic and paragraph structure of this review are not very satisfactory. I suggest that the author make major revisions to the entire text.
Why does the author emphasize climate change in the title?
The species name should be written in full the first time it appears, but can be abbreviated upon subsequent mentions. For example: Alligator mississippiensis, A. mississippiensis.
For crocodilian species where TSD has been confirmed, images of these crocodiles could be included for illustration.
The formatting of the paper is quite poor.
Line19: egg? Eggs have gender? I think “embryo” is right. Or like this: The gender of crocodilians is determined by the temperature to which their eggs are exposed during incubation.
Lin27-29: The impact of climate change is not well represented here. I suggest making modifications.
Line48-51: The relevant references are missing.
Lin53-54: Your focus should be entirely on crocodiles. Not other reptiles.
Line 58: In section 2. Materials and Methods, you should describe the specific process of screening. Figure 1 is only a visual representation of your screening process and should not be solely relied upon by the readers. Some specific factors for excluding references might not be adequately represented in Figure 1.
Line73-74: I do not believe that all reptiles lack sex chromosomes.
Line100-116: I suggest creating a table summarizing the relationship between incubation temperature and sex ratio in different crocodilian species as reported in references.
Line146: In Figure 2, photos of the species could be added to make it clearer.
Line355: I suggest presenting the molecular mechanism of TSD in crocodiles in the form of a table. The table can list the genes that have been studied so far, the species they have been studied in, and whether they are upregulated or downregulated at different temperatures.
Author Response
Dear Sir/Madam,
thank you for your review. Please see attached our response to your comments and suggestions.
Kind regards

Round 2
Reviewer 3 Report
Comments and Suggestions for Authors
I think this revision could be accepted.